# Lymph Node Reporting and Data System (LN-RADS)—Retrospective Evaluation for Ultrasound Classification of Superficial Lymph Nodes

**DOI:** 10.3390/cancers17122030

**Published:** 2025-06-18

**Authors:** Cezary Chudobiński, Katarzyna Pasicz, Małgorzata Hanke, Adam Kaczmarek, Mateusz Pajdziński, Agnieszka Kołacińska-Wow, Leszek Gottwald, Wojciech Kuncman, Michał Podgórski, Andrzej Cieszanowski

**Affiliations:** 1Department of Radiology, Copernicus Memorial Hospital, 62 Pabianicka Str., 93-513 Lodz, Poland; cezary.chudobinski@wp.pl (C.C.); malgosiahanke@gmail.com (M.H.); kaczmarek.umed@gmail.com (A.K.); 2Medical Physics Department, The Maria Sklodowska-Curie National Research Institute of Oncology, 15/B Wawelska Str., 00-001 Warsaw, Poland; katarzyna.pasicz@nio.gov.pl; 3Department of Teleradiotherapy, Regional Cancer Centre, Copernicus Memorial Hospital of Lodz, 62 Pabianicka Str., 93-513 Lodz, Poland; mateusz.pajdzinski@gmail.com; 4Department of Oncological Physiotherapy, Medical University of Lodz, 62 Pabianicka Str., 93-513 Lodz, Poland; agnieszka.kolacinska@umed.lodz.pl; 5Department of Radiotherapy, Chair of Oncology, Medical University of Lodz, 62 Pabianicka Str., 93-513 Lodz, Poland; lgottwald@wp.pl; 6Department of Pathology, Chair of Oncology, Medical University of Lodz, 251 Pomorska Str., 92-213 Lodz, Poland; wojciechkuncman@gmail.com; 7III Department of Radiology, Medical University of Lodz, 113 Żeromskiego Str., 90-549 Lodz, Poland; 8Department of Radiology I, The Maria Sklodowska-Curie National Research Institute of Oncology, 15/B Wawelska Str., 00-001 Warsaw, Poland; acieszanowski@wum.edu.pl; 92nd Department of Clinical Radiology, Medical University of Warsaw, 1ABanacha Str., 02-097 Warsaw, Poland

**Keywords:** lymph nodes, ultrasound, neoplasms, metastasis

## Abstract

Nodes play a key role in detecting and monitoring cancer, but doctors currently lack a clear and standardized way to describe what they see in imaging tests. To help with this, we created a system, called LN-RADS, that allows radiologists to evaluate and classify lymph nodes based on how suspicious they look. In this study, we tested LN-RADS by analyzing nearly 720 lymph nodes from almost 500 patients and comparing the results to actual tissue samples. Our system proved to be accurate and reliable in identifying cancerous lymph nodes and in estimating the level of cancer risk. One feature, the thickness of the outer layer of the lymph node, turned out to be especially helpful in predicting whether cancer was present. We believe this system can improve communication between radiologists and other doctors, support more consistent diagnoses, and ultimately help guide better treatment decisions for patients.

## 1. Introduction

The diagnosis of LNs malignancy is a critical component of oncology at every stage of the disease—staging, treatment, and follow-up—and has a profound impact on overall outcomes. Nguyen et al. performed a retrospective cohort analysis of over 1.3 million patients [1] and presented three key findings relevant to clinical oncology:Significant impact of nodal burden on survival: The study highlights that the number of affected LNs is a critical predictor of mortality across various solid cancers. An increased nodal burden is consistently associated with poorer survival outcomes, underscoring the importance of comprehensive nodal evaluation in cancer staging and prognosis.Nodal burden as a superior prognostic factor: In certain cancers, nodal burden was found to be a stronger predictor of patient outcomes than traditional prognostic factors, such as tumour size. This suggests that the extent of LN involvement may be more indicative of disease progression and survival than other commonly used metrics.Implications for personalized therapy: The findings suggest that a detailed assessment of LN involvement could inform more personalized treatment strategies. By quantifying nodal burden, clinicians may better stratify patients according to risk and tailor therapies to improve survival outcomes.

Numerous other studies, focusing on specific types of cancers or body regions, have consistently demonstrated a strong correlation between the extent of LN involvement and treatment outcomes. The prognostic significance includes the number of involved lymph nodes, the size of metastatic lesions within the nodes, the presence of extranodal extension, and the number of nodal levels affected. The authors suggest that a thorough morphological evaluation of the lymph nodes and more personalized treatment approaches could improve patient survival outcomes [2,3,4,5,6,7,8,9].

If the role of lymph nodes is so immense and treatment outcomes remain far from perfect, the question arises as to whether current methods of lymph node assessment fully utilize the technical capabilities of modern radiology.

Among the many criteria used for evaluating LNs, size, especially the short-axis diameter (SAD), is a frequently used criterion. It is known that reactive LNs are usually enlarged, whereas malignant LNs in the early stages are small. There is no single ideal cutoff point, such as the SAD, that could be universally applied across different clinical scenarios. Depending on the researcher, the region of the body, or cancer type, various SAD cut-off points close to 10 mm have been reported, typically ranging between 8 and 12 mm [10,11,12,13,14,15,16,17,18,19,20]. Relying on a single parameter has several significant drawbacks and must be a compromise between sensitivity and specificity.

The adoption of a single fixed criterion, such as the commonly cited 10 mm SAD, introduces significant errors across all stages of cancer management:Staging Phase: Relying on a rigid size criterion may result in an underestimated stage of disease, leading to a limited surgical scope, insufficiently aggressive systemic therapy, or even the omission of adjuvant therapies, such as radiotherapy, increasing the risk of disease recurrence.Treatment Phase: Neglecting macrometastases (Figure 1) during active treatment can delay necessary interventions by weeks or even months, allowing for the potential spread of cancer to an extent that may evade control and treatment.Post-Treatment Surveillance: Similarly, during follow-up, there is a risk of overlooking macrometastases despite the technical capability to detect them, which may hinder timely therapeutic interventions and negatively impact patient prognosis.

Relying on a single criterion also becomes a problem when applying a fixed cutoff point across all LNs in various regions of the body or different cancer types. For example, in the neck region, numerous studies have demonstrated that lymph nodes in Groups I and II are naturally larger than those in other cervical regions, requiring a higher cutoff threshold for these nodes [21]. Despite the same region, the threshold can be dependent on the tumor biology. Lymph node metastases in prostate cancer are present in 70% of LNs < 8 mm, whereas bladder cancer metastases are present in large LNs (75% of LNs > 8 mm [10]). There are additional environmental factors that can impact the lymph node size, such as a history of vaccination or infections (e.g., chronic leg ulcers with recurrent erysipelas), which can mimic metastases and should be considered [22]. The criteria for qualifying a patient with tonsillitis for an LN biopsy differ significantly from those for a patient with advanced pharyngeal cancer. In the case of pharyngeal cancer, even when LNs are clinically assessed as N0, the risk of metastasis remains high, warranting the use of a lower cutoff point for biopsy, thereby including even small, suspicious nodes. This illustrates the difficulty in applying rigid, uniform criteria across broader clinical practice. A more flexible, individualized approach is required, tailored to the specific clinical scenario.

Many studies have suggested that relying on a single criterion—especially size—may not be optimal for assessing the nature of a lymph node. Several other criteria have been proposed, including the assessment of shape, echogenicity, homogeneity, vascularization, elasticity, density, enhancement pattern, and signal characteristics on MRI, especially diffusion-weighted imaging (DWI) and the apparent diffusion coefficient (ADC). Some authors have proposed systems combining the evaluation of multiple parameters to increase both the sensitivity and specificity of the diagnosis of LNs malignancy. Ryu et al. [23] developed the Cervical Lymph Node Imaging Reporting and Data System (CLN-RADS), using ultrasound as a structured reporting system. After the retrospective analysis of 291 patients with cervical lymphadenopathy, they concluded that CLN-RADS is reproducible, with a relatively high degree of interobserver agreement. In 2019, Chung et al. [24], in a retrospective study including 191 patients, proposed a 9-point system using CT to stratify the risk of LN metastasis exclusively in patients with head and neck squamous cell carcinoma (HNSCC). They suggested that this system can facilitate decision making for patients with HNSCC in daily practice. Ni He developed a scoring system for diagnosing malignant axillary lymph nodes on MRI, achieving a high specificity of 91% and a sensitivity of 93% [25]. In 2021, Elsholtz et al. [26] introduced the Node Reporting and Data System 1.0 (Node-RADS) for the standardized reporting of possible cancer involvement of regional and distant LNs on CT and MRI at any anatomical site. The problem is that each of these systems is entirely different, addressing a specific body region, type of cancer, or imaging modality. Each one requires extensive data for scoring, and in the absence of some data, it ceases to function correctly. Furthermore, scoring points are often assigned arbitrarily, as integers, and may not reflect the true proportions of predictive strength for specific features. This strength may also change depending on other factors. A notable drawback of such “calculation-based” scoring systems is the need to sum numerous points and mentally create scoring groups, which is time-consuming and can lead to calculation errors. The variety of systems requires radiologists and clinicians to learn and efficiently apply each one, which can be an additional challenge. In our opinion, the solution to these inconveniences could be a single, universal, intuitive system based on a fast heuristic multiparametric assessment, which is both quick to evaluate and easy to communicate between the radiologist and clinician. In this article, we present the LN-RADS scale in the context of ultrasounds, which may be useful in various clinical situations, especially for the assessment of superficial LNs. The system aims to detect small metastases early based on multiparametric assessment, rather than relying on a single criterion. Ultrasound, due to its excellent spatial resolution, enables a detailed assessment of several features of LNs, such as SAD, long axis diameter (LAD), shape, margins, structure, echogenicity, vascularity, and elasticity. Since some clinicians may not be fully familiar with the complexities of radiological assessment of various LNs characteristics, the system provides a comprehensive interpretation of all available parameters by assigning a category within the LN-RADS scale, which greatly enhances communication between radiologists and clinicians. Moreover, the risk stratification for malignancy across specific categories enables the selection of optimal management strategies.

## 2. Materials and Methods

### 2.1. Patient Population and LNs’ Characteristics

This is a retrospective, single-center study. For the initial assessment, all ultrasound-guided biopsies of superficial LNs performed between June 2015 and August 2021 were included. Only cases with conclusive histopathology and digitally recorded images enabling the assessment of LN-RADS criteria were retained for analysis. Figure 2 presents a flowchart documenting inclusion of cases into the final analysis and reasons for exclusion.

The total number of eligible cases was 719 LNs from 489 patients, which were included in the final analysis (172 men, 317 women, aged 19–93, mean age of 59). In all cases, B-mode US was available, and in 238 cases, Color Doppler US was also available. Samples were obtained from superficial LNs of the neck (*n* = 243), peri-clavicular region (*n* = 51), axilla (*n* = 278), and inguinal region (*n* = 147). All observations were compared with the histopathology results (461 fine needle biopsies [FNB], 237 core needle biopsies [CNB], 20 surgical biopsies, and 1 vacuum-assisted biopsy [VAB]).

### 2.2. Imaging Data Analysis

Figure 1 features a flowchart illustrating the selection process of LN cases for the study. From an initial set of 1029 LNs, exclusions were made based on image availability, quality, and histopathological (HP) diagnostic criteria, resulting in 719 eligible cases.

All US studies of 719 LNs were performed by senior consultant radiologists experienced in oncology imaging and biopsies under US guidance. The Esaote My Lab Twice system, with a high-resolution probe (LA533 3–13 MHz), was used. The images of LNs were transferred in TIFF format to the external hardware and evaluated independently by three radiologists: a senior consultant with 20 years of experience who performed studies (A), a radiology consultant with 18 years of experience (B), and a fourth-year resident (C) who underwent a 3-month internship in the oncologic radiology department, where he became familiar with ultrasound images of nodes and the LN-RADS scale.

### 2.3. LN-RADS Assessment Guidelines

The Lymph Node Reporting and Data System (LN-RADS) is a structured reporting protocol developed to categorize LNs based on their imaging characteristics and estimated malignancy risk, with the intent of standardizing diagnoses [27]. This system is similar to BI-RADS for breast imaging and aims to facilitate communication and decision making among medical professionals. The scale classifies LNs into six groups: 1, 2, 3, 4a, 4b, and 5, each corresponding to specific imaging features and associated malignancy risk.

The LN-RADS assessment is based on general, flexible criteria and fast heuristic evaluation. The system is open and multiparametric, meaning that it can incorporate a variety of radiological features as well as additional available clinical information. This includes the patient’s medical history, e.g., tumor-related or inflammatory conditions, general condition, laboratory results, and available morphological features of the LNs. The principles of LN-RADS were created based on numerous scientific articles and the author’s own experience. Given the complexity of LNs evaluation, there is no simple equation that can be applied to calculate the risk of malignancy. Therefore, the presented data should be viewed as guidelines and applied with flexibility in clinical practice. In cases where multiple clinical and morphological elements are present, the overall estimated probability of malignancy should be considered.

One of the premises of this scientific work is the stratification of lymph node malignancy risk into groups, which aims to help the radiologist select the appropriate category for a given node and, on the other hand, assist clinicians in making decisions, such as whether to perform a biopsy. LN-RADS categories are as follows:

Groups 1 and 2 are categorized as benign nodes with no concerns.

Group 3 includes reactive benign nodes, but there is a risk of misdiagnosis, especially in systemic diseases (e.g., leukemia or lymphoma) or metastatic nodes with regular cortical thickening mimicking reactive changes.

Group 4a consists of nodes slightly suspicious for malignancy.

Group 4b includes nodes highly suspicious for malignancy.

Group 5 contains nodes with obvious malignant features.

LN-RADS categories according to radiological features and clinical data:LN-RADS 1: Normal. No enlargement (recommended max SAD up to 6–7 mm), oval shape (L/S-ratio > 2), regular cortex with maximum thickness ≤ 3 mm, cortex echogenicity similar or higher to the background fatty tissue, smooth margins, no other changes in architecture (no calcifications, no fluid collections, no necrosis, and no FCT [focal cortical thickening] or LCT [local cortical thickening]), and no pathological peripheral or chaotic vascularizationLN-RADS 2: Steatotic. LNs can be enlarged in one or both axes, with regular cortex with a maximum thickness of ≤3 mm, hilum hyperechoic (steatotic) with no size limits, no other changes in architecture (no calcifications, no fluid collections, no necrosis, no FCT, and no LCT), and no pathological peripheral or chaotic vascularization.LN-RADS 3: Reactive. Probably due to an inflammatory process or vaccination. Dominant feature: thickened cortex > 3 mm, regular or with discrete irregularity, general enlargement in one or two axes, preserved oval shape (L/S-ratio > 2), preserved medulla, no other changes in architecture (no calcifications, no fluid collections, no necrosis, and no focal cortical thickening (FCT)), cortex echogenicity similar to or moderately lower than the background fatty tissue, well-defined margins, no pathologic peripheral or chaotic vascularization (small vessels in hilum can be seen), no oncological or haematological history, and no laboratory oncological abnormalities.LN-RADS 4: Suspicious for malignancy, thus LNs that morphologically do not match group 1, 2, 3, or 5 or have additional radiological or clinical factors increasing probability of malignancy in LNs categorized as LN-RADS 3, i.e., high or increasing laboratory markers (i.e., PSA for inguinal LNs); active neoplasm in the region (i.e., breast cancer for axillary LNs); another metastatic or systemic LN in the region; and clinical symptoms suggesting oncological or systemic hematological disease. The main rule of selecting LNs for group 4 is “better check than miss”. The LN-RADS 4 category is divided into two subcategories:
-4a: low suspicion for malignancy—size may be normal in SAD and LAD, cortex with thickening over 3 mm, and moderate irregularity, especially LCT. It is assumed that all 4a LNs should be verified by biopsy or PET. If biopsy is not possible, they should be treated as suspected and malignant.-4b: high suspicion of malignancy—size may be normal in SAD and LAD; cortex thickening over 4 mm and irregularity, especially FCT, or no hilum; shape is more round than oval (L/S-ratio ≤ 2); hypoechogenicity to background fatty tissue, especially nearly anechoic “black hole sign”; micro-calcifications; fluid collections; necrosis; abnormal peripheral or chaotic vascularization architecture; ill-defined/blurred margins.
LN-RADS 5: Definitely malignant. Enlargement in SAD and more malignancy features: cortex thickening over 6 mm, lack of hilum, hypoechogenicity to the background fatty tissue or “black hole sign”, evident cortex irregularity (LCT and FCT), shape more round than oval (L/S-ratio ≤ 2), micro-calcifications, fluid collections, necrosis, abnormal peripheral or chaotic vascularization architecture, ill-defined/blurred borders, or signs of extracapsular infiltration.

### 2.4. The Assessment Based on the LN-RADS Scale

In the first stage, three radiologists, using the guidelines for LN-RADS, evaluated all 719 LNs and categorized each into one of six categories.

In the second stage, the images of 719 LNs were further evaluated by an expert reader for 12 features: 6 objective and 6 subjective morphological features: SAD (short-axis diameter), LAD (long-axis diameter), CTD (cortical thickness diameter), MTD (medullary thickness diameter), SAD/LAD, MTD/(MDT + CTD) ratio (Figure 3), shape, cortex homogeneity, cortex irregularity, margins, cortex echogenicity, and vascular architecture (Table 1). The values of each feature as predictors of node malignancy were determined.

### 2.5. Statistical Analysis

Statistical calculations were performed using R environment version 3.6.3 with appropriate packages [28,29,30] and Jamovi version 1.6. Differences at the *p* < 0.05 level were considered significant. The normality of the data was checked using the Shapiro–Wilk test. Differences between malignant and benign LNs were tested using the Mann–Whitney Test, while differences in LNs’ features between neoplasms were tested with the Kruskal–Wallis ANOVA, with post hoc tests. Comparisons of categorical data were performed using the Chi-squared test. The AUC with a 95% confidence interval, sensitivity, specificity, PPV, NPV, accuracy, and threshold value were determined. The cut-off points were determined by maximizing the Youden Index (Youden’s J-statistic). In the aforementioned statistics, parameters were derived only from the senior consultant radiologist. Cohen’s kappa statistic test and percent agreement were used to test for inter-rater reliability. The kappa values were interpreted as follows: slight agreement: 0.01–0.20; fair agreement: 0.21–0.40, moderate agreement: 0.41–0.60, substantial agreement: 0.61–0.80, and almost perfect agreement: 0.81–1.00 [30].

## 3. Results

### 3.1. Assessment of the Accuracy of LN-RADS for Differentiation Between Malignant and Benign Superficial LNs in US

The LN-RADS system falsely classified 92 (13%) LNs (49 false-positive and 43 false-negative diagnoses; Table 2), yielding 89% (95% CI 85–92%) sensitivity, 85% (95% CI 81–88%) specificity, 87% (95% CI 84–89%) accuracy, 88% (95% CI 84–90%) positive predictive value, and 86% (95% CI 83–89%) negative predictive value for the diagnosis of malignant lymph nodes.

It should be underlined that the key rule of group LN-RADS-4 is “better check than miss”. Therefore, category LN-RADS-4a (suspicious, but with a low probability of malignancy) is, in terms of statistics, treated as rather benign, but according to the LN-RADS algorithm, these LNs still need further investigation—biopsy or PET-CT. Therefore, in daily practice, the number of missed malignant LNs is minimized (from 43 to only 2 cases in our study).

### 3.2. Assessment of the Risk of Malignancy in Each Group of LN-RADS

LN-RADS classification was compared with histopathologic data, and the risk of malignancy in each category was determined as shown below (Figure 4):LN-RADS-1—33 normal LNs—0% risk of malignancy.LN-RADS-2—46 steatotic LNs—0% risk of malignancy.LN-RADS-3—109 reactive LNs—2% risk of malignancy.LN-RADS-4—320 LNs with suspicion of malignancy further divided into the following categories:

a. LN-RADS-4a—136 LNs with a lower risk of malignancy, reaching 31%.

b. LN-RADS-4a—184 LNs with a higher risk of malignancy, reaching 77%.

5.LN-RADS-5—211 definitely malignant LNs, with a 97% risk of malignancy.

### 3.3. The Assessment of the Agreement Between the Readers

#### Inter-Observer Agreement

Cohen’s kappa value between expert reader (A) and experienced reader (B) was 0.766 (95% CI: 0.729–0.802), that between expert reader (A) and trained resident (C) was 0.686 (95% CI: 0.647–0.726), and that between experienced reader (B) and resident (C) was 0.655 (95% CI: 0.614–0.695). Interobserver agreement in all analyzed cases was substantial.

### 3.4. The Assessment of Morphological Features Allows for Identifying the Predictors of Benign or Malignant LNs—Data Are Presented in Table 3 and Figure 5

#### Cohorts Analysis

LN-RADS was designed as a universal risk stratification system. However, we assessed whether the final sensitivity results were influenced by the type of neoplasm that metastasized. A total of 389 neoplastic LNs were categorized into four cohorts with the following sensitivity outcomes:233 LNs with cancer—27 LNs classified as 4a, 99 as 4b, and 107 as 5, yielding a sensitivity of 88%.93 LNs with leukemias/lymphoma—1 LN classified as 3, 8 as 4a, 22 as 4b, and 62 as 5, yielding a sensitivity of 90%.48 LNs with melanoma/sarcoma—3 LNs classified as 4a, 15 as 4b, and 30 as 5, yielding a sensitivity of 94%.17 nonspecific neoplastic LNs—6 LNs classified as 4a, 6 as 4b, and 5 as 5, yielding a sensitivity of 64%.

It is worth mentioning that in the above calculations, LNs from group 4a were assumed to be more likely to be benign than malignant (30% estimated probability of malignant ca). Nevertheless, according to our recommendation, they would require further diagnostics/pathomorphological verification. Thus, the actual sensitivity would be higher.

Additionally, the morphological parameters used for LN-RADS classification were analyzed and are presented in Table 4.

**Table 3 cancers-17-02030-t003:** Sensitivities, specificities, accuracies, positive predictive values, and negative predictive values for the diagnosis of malignant lymph node for separately analyzed parameters.

Parameter	Threshold Value	Sensitivity	Specificity	Accuracy	PPV	NPV
LAD	17 mm	55%	58%	57%	61%	52%
SAD	9 mm	69%	75%	72%	76%	67%
S/L ratio	0.51	80%	62%	72%	71%	73%
CTD	6 mm	89%	74%	82%	80%	85%
MTD/(MTD + CTD) ratio	0.24	78%	78%	78%	81%	75%
Cortex echogenicity *	(-) *	68%	82%	74%	81%	68%
Cortex irregularity *	(-) *	90%	68%	80%	77%	86%
Margins *	(-) *	8%	98%	49%	83%	47%
Inhomogeneity *	(-) *	25%	92%	56%	79%	51%
Shape *	(-) *	75%	76%	75%	79%	72%
Vascular architecture **	(-) *	77%	70%	74%	84%	60%
Total LN-RADS		89%	85%	87%	88%	86%

* categorical variables; ** vascular architecture was analyzed in 238 lymph nodes studied with color Doppler. LAD—long-axis diameter; SAD—short-axis diameter; CTD—cortical thickness diameter; MTD—medullar thickness diameter. (-) no paramter that can be measured for this parameter.

**Table 4 cancers-17-02030-t004:** Comparison of morphological parameters between LN cohorts. Values are presented as mean (standard deviation).

Feature	Cancers [mm]	Leukemias/ Lymphoma [mm]	Melanomas /Sarcoma [mm]	Undifferentiated [mm]	Benign [mm]	*p*
LAD	17.2 (10.0)	29.9 (11.7)	23.3 (14.7)	16.6 (9.2)	17.2 (9.1)	0.0001
SAD	12.0 (7.7)	18.2 (7.6)	15.8 (11.6)	10.0 (6.9)	7.4 (3.3)	0.0001
CTD	11.5 (7.9)	15.4 (8.4)	15.8 (11.6)	10.1 (7.1)	4.5 (3.2)	0.0001
MTD	0.9 (1.8)	2.0 (2.5)	0.9 (1.9)	1.1 (1.6)	2.9 (2.4)	0.0001

LAD—long-axis diameter; SAD—short-axis diameter; CTD—cortical thickness diameter; MTD—medullar thickness diameter. According to post hoc analysis for all morphological parameters, significant differences were found between LNs with cancers and those with leukemias/lymphomas.

**Figure 5 cancers-17-02030-f005:**
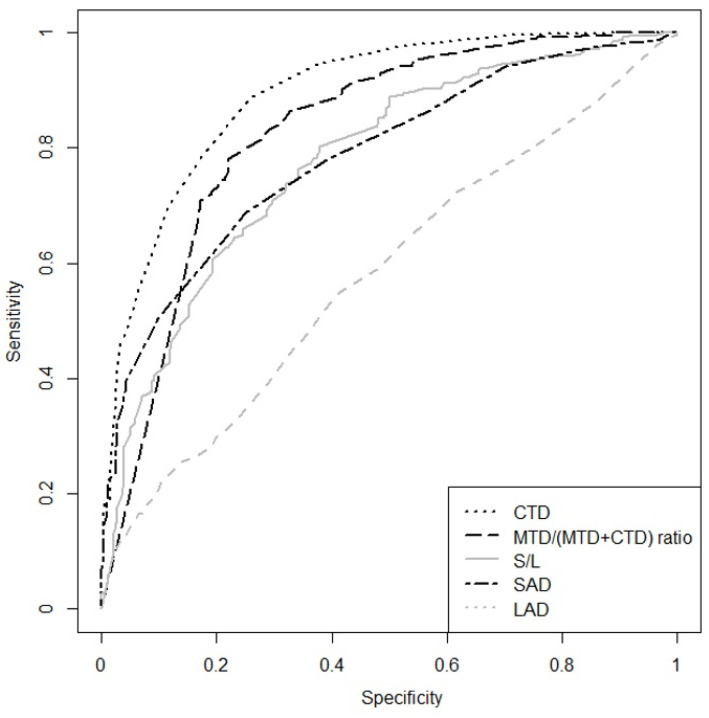
Receiver operating characteristic (ROC) curve for LAD (AUC = 0.582; 0.541–0.624), SAD (AUC = 0.785; 0.753–0.818), S/L (AUC = 0.777; 0.743–0.811), CTD (AUC = 0.894; 0.871–0.917), and MTD/(MTD + CTD) ratio (AUC = 0.828; 0.799, 0858) for differentiation between benign and malignant lymph nodes.

## 4. Discussion

In the search for an optimal method of assessing lymph nodes, we analyzed various assessment models, among which three dominant models emerged, each presenting a different philosophy:The single rigid criterion model, e.g., 10 mm SAD;The multiparametric model with strictly defined features and rigid cutoff values—the so-called calculation system, e.g., Node RADS;The open, flexible system utilizing a wide range of available radiological and clinical data based on quick heuristic evaluation—e.g., LN-RADS.

This study evaluated the LN-RADS scale as a tool for assessing superficial LNs and facilitating communication between radiologists and clinicians. Given the complexity of the structural assessment of LNs and the need to consider various data depending on individual circumstances, a decision was made to create an open multiparametric model based on fast heuristic evaluation, similar to BIRADS. This approach was preferred over using a single parametric model, such as the commonly applied 10 mm short-axis diameter (SAD) criterion, or a closed model based on a limited number of parametric calculations, like Node-RADS. This approach allows for the efficient assessment of multiple morphological features while also considering clinical data. The flexible assessment of multiple parameters enables the utilization of predictors, some of which exhibit high sensitivity and others high specificity. This means that we can compensate for the weaknesses inherent in individual assessments and eliminate the undesirable “sieve effect”, where one must choose between high sensitivity or high specificity or accept a compromise that often falls short of expectations—resulting in a significant number of false positives (FP) and false negatives (FN). By leveraging this multiparametric, flexible evaluation, the LN-RADS scale achieves high diagnostic accuracy and effectively captures specific, commonly encountered patterns of lymph nodes, correlating them with defined levels of malignancy risk.

It is evident that during the initial evaluation of LNs, the underlying pathology is often unknown—whether the nodes are reactive, lymphoma, leukemia, or metastatic carcinoma. Furthermore, a subset of patients may present with two or more concurrent malignancies. Therefore, an effective diagnostic system must operate independently of the type of primary pathological process.

An analysis of our study cohort demonstrates that the LN-RADS system proved effective across a wide range of malignancies. Thanks to its open multiparametric assessment and the absence of rigid criteria, the flexible LN-RADS framework facilitated the detection of malignant LNs in not only diverse types of carcinomas but also hematologic malignancies. As is well-known, some LNs in leukemias and lymphomas may mimic reactive nodes; nevertheless, LN-RADS demonstrated high diagnostic accuracy across the entire study and within individual subgroups.

A comparison between the LN-RADS system and the 10 mm size criterion revealed that LN-RADS identified 115 malignant lymph nodes that were metastatic and smaller than 10 mm in SAD, accounting for 22% of the entire study cohort. This finding indicates that nearly one in four LNs in the analyzed study would have been falsely categorized as negative if the 10 mm SAD criterion had been applied. This sheds light on the potential scale of medical errors resulting from inadequate lymph node assessment at various stages of oncological care, from initial staging through treatment to follow-up. It can be hypothesized that the efficacy of many modern therapies is constrained by the suboptimal detection of metastatic changes in LNs.

It is difficult to directly compare the LN-RADS and Node-RADS systems, as our evaluated study is based on ultrasound studies, while the Node-RADS scale was developed for CT and MRI. However, it is possible to compare the differing structures of the systems, particularly in selected aspects that seem to highlight weaknesses in the calculation-based system. Unlike LN-RADS, the Node-RADS calculation system is based on rigid cutoff points and arbitrarily assigned score values for the assessed features, which are integers and do not reflect the true predictive strength of the given feature. The system does not allow for the inclusion of additional strong predictors, such as cortex thickness or the presence of FCT. It also requires memory-based calculation of the total score and categorization into a specific group, with node classification only possible after determining the final range, which seems time-consuming and increases the risk of error. Additionally, there are no assumptions regarding the estimated risk of cancer at each stage of the scale, which may lead to significant differences in assessment between researchers and, consequently, prevent the use of results for clinical decision making due to the lack of consistency in evaluations. As seen, the philosophy of the open heuristic system, such as LN-RADS, differs significantly from the calculation-based system exemplified by Node-RADS. However, to objectively assess their value, further studies of both systems in CT and MRI images are necessary.

### 4.1. Analysis of Features as Predictors

In the second part of our study, we analyzed 12 morphological features of lymph nodes to identify predictors of malignancy or benignity (Table 1). Some features demonstrated a strong positive predictive value, while others were more indicative of benignity, which is consistent with our previous observations and the foundational principles of the LN-RADS. Leveraging this knowledge through multiparametric assessment allows for significantly improved differentiation between benign and malignant lymph nodes compared to single-criterion systems, which are prone to the limitations of a “sieve effect.” We compared our findings with observations from numerous other scientific publications.

Short-axis diameter (SAD)**.** Among several features used for discrimination between malignant and benign LNs, the SAD is the most commonly used. Despite known drawbacks, such as frequent false-positive as well as false-negative diagnoses, this parameter was applied to Response Evaluation Criteria in Solid Tumors (RECISTs) version 1.1, defining LNs ≥10 mm in the SAD as pathologically enlarged [20]. It is important to emphasize that although RECIST was originally designed to assess treatment response rather than to differentiate benign from malignant lymph nodes, its simplicity and widespread use in radiology and oncology have led to its application as a criterion for distinguishing between healthy and cancer-affected nodes. Regardless of the 10 mm criterion in RECIST, numerous publications also reference a 10 mm threshold as differentiating benign from malignant nodes [10,11,12,13,14,15,16,17,18,19,20,31,32,33]. Interestingly, in our study of 719 LNs, the optimal threshold value for discrimination of malignant and benign nodes was ≥9 mm. Separate analysis performed for both threshold values revealed a slightly higher accuracy of 72% (69% sensitivity, 75% specificity, 204 false results: 83 FP and 121 FN) for the 9 mm cut-off value than for the threshold of 10 mm (70% accuracy, 61% sensitivity, 81% specificity, 213 false results: 63 FP and 150 FN). Consequently, the 10 mm threshold led to a higher number of false-negative diagnoses. Since there are several cancers that commonly metastasize to small regional LNs (e.g., prostate, rectal, and pancreatic cancers), the use of a 10 mm SAD threshold may lead to understating these patients and neglecting the chance for appropriate treatment. In the context of the SAD, LN-RADS enabled a significant reduction in the number of false-negative results and reduced the undertreatment phenomenon. Interestingly, it similarly lowers the number of false-positive results, which often lead to overtreatment of benign LNs. Comparing two different approaches—LN-RADS classification and traditional 10 mm SAD, we obtained much higher accuracy for LN-RADS, 87%, than the 10 mm SAD, 70%, in the diagnosis of malignant LNs.

The long-axis diameter (LAD), in line with findings from other studies, has proven to be a weak parameter for differentiating lymph nodes, with an AUC of 0.582 (95% CI: 0.541–0.624). Despite this limitation, a 15 mm cutoff for LAD has been included in the Cheson criteria since 2017 [34]. In the case of the International Working Group consensus response evaluation criteria in lymphoma (RECIL 2017), lymph nodes can be considered target lesions if their longest diameter measures ≥15 mm. In this context, the 15 mm LAD threshold is only relevant for selecting target lymph nodes, while treatment response evaluation relies on the product of LAD × SAD. Such a surface-based parameter allows for a reliable assessment of treatment effectiveness. It is crucial to emphasize that measuring the LAD alone does not hold predictive value for malignancy. Moreover, in the case of a radiological report, if it is not explicitly stated that the isolated measurement specifically refers to the long-axis diameter (LAD), it may be mistakenly interpreted as the short-axis diameter (SAD), potentially leading to clinical errors. Therefore, lymph node dimensions should always be reported as a product of LAD × SAD (e.g., 12 × 6 mm) or as the SAD alone with appropriate annotation (e.g., 9 mm SAD).

S/L ratio—the analysis of the S/L ratio as a measurable parameter and the shape of the lymph node as a subjective feature revealed that slender, elongated, and narrow nodes are usually benign, whereas the more rounded the shape, the higher the risk of malignancy, with a cutoff point of approximately 0,5.

Cortical thickness diameter (CTD). Among the 12 analyzed morphological features of lymph nodes, cortical thickness emerged as the strongest predictor of malignancy; however, it yielded a lower accuracy of 82% compared to the entire multiparametric assessment with LN-RADS (87%). Choi et al. showed that a CTD greater than 3 mm was the most accurate indicator of malignancy, with a 4.14 times increased risk of the presence of an axillary LNs metastasis compared to a cortical thickness less than 3 mm [34]. In our study, the CTD was also the most reliable solitary factor of metastatic LNs; however, according to AUC analysis, the optimal cut-off value for the diagnosis of malignancy was higher (6 mm), yielding an accuracy of 82%.

The MTD/(MTD + CTD) ratio, representing the proportion of the cortical thickness to sinus thickness, proved to be a valuable predictor of malignancy. A developing metastasis often thickens the cortex while simultaneously compressing or infiltrating the sinus, leading to its narrowing. An increasing cortical thickness relative to the sinus signals a higher risk of malignancy in the lymph node. This parameter is particularly valuable in small nodes, as it is used to detect relatively slight cortical thickening or sinus narrowing that disrupts the normal structural proportions of the node, even when these absolute changes do not exceed discriminatory threshold values. MTD, as an independent predictor of malignancy, has a significance similar to the subjective presence or absence of a visible hilum. The smaller the MTD value, particularly a value of 0 (absence of hilum), the more it indicates an increased risk of a neoplastic process. However, considering the natural differences in lymph node size and hilum dimensions, the isolated MTD parameter has less significance than the MTD/(MTD + CTD) ratio, indicating potential disruption of the cortex-to-hilum thickness proportion. Similar conclusions were reached by Song in his study, demonstrating the high value of the cortex-to-medulla ratio in lymph nodes [35].

Cortex irregularity. We noticed two important patterns of LNs’ metastatic invasion—focal cortical thickening (FCT) and local cortical thickening (LCT). The FCT is usually the result of the centrifugal growth of cancer cell nets forming macro-metastases—in the early stage, small, isolated round foci grow, and in more advanced stages, FCTs grow, merge, and successively infiltrate the entire LN. The LCT is more subtle cortex thickening and may be dependent on malignancy or inflammatory processes. In the same study, Choi et al. demonstrated that the absence of a hilum exhibited the highest specificity for detecting axillary lymph node metastasis in breast cancer [34]. Our findings suggest that the absence of a sinus is a late sign indicative of an advanced stage of malignancy within the lymph node (Fig. evolution of metastasis). We recommend including features such as the LCT, and especially the FCT, in lymph node assessment, as these support significantly earlier detection of metastases. The majority of authors agree that the most important features of benign or reactive LNs on B-mode US are a well-preserved hyperechoic hilum; oval shape; longest to transverse diameter ratio >2; and thin, regular cortex. In contrast, metastatic LNs are more rounded, irregular, and without a hilar echo, but they have a thick, irregular cortex [34,35,36]. Our results confirmed the majority of these observations.

The echogenicity of the lymph node cortex is significant in detecting malignancy. Typically, hyperechoic nodes are post-inflammatory, fibrotic, and fatty, with a very low risk of malignancy. However, an exception may be metastases from thyroid carcinoma and head and neck squamous cell carcinoma (HNSCC). In the case of thyroid cancer metastases, the histological structure may resemble that of the thyroid or contain microcalcifications, resulting in higher echogenicity. In HNSCC, hyperechogenicity is caused by tissue inhomogeneity associated with microfoci of necrosis, especially after treatment. Moderate echogenicity of the cortex, similar to that of fatty tissue, typically characterizes non-malignant lymph nodes. Low echogenicity, particularly anechoic lymph nodes (the “black hole sign”), correlates with malignant processes in lymph nodes [34,36,37,38]. Sometimes, the nodes are completely anechoic, so they initially resemble fluid-filled cysts. This phenomenon can be observed in both carcinomas and lymphomas, as well as leukemia. In lymphomas and leukemia, a characteristic delicate reticular pattern can be observed against the pure anechoic cortex. Although echogenicity is a subjective parameter, numerous publications, as well as our own work, have shown that low echogenicity of a lymph node is a strong predictor of malignancy [39,40,41].

The homogeneity functions as a characteristic independently of echogenicity; however, these terms are sometimes confused. Homogeneity is an interesting and atypical feature in the context of small neoplastic changes, i.e., macro-metastases, as it is size-dependent. Generally, inhomogeneity of lymph nodes is widely regarded as a feature indicative of malignancy, while homogeneity is typically associated with benignity. Our study revealed that this is somewhat of a cognitive trap, as this holds true only for larger lymph nodes, where insufficient tumour angiogenesis leads to inadequate vascularization, resulting in necrosis and inhomogeneity. In the case of smaller neoplastic foci (macro-metastases), necrosis is very rare—almost all small metastatic lymph nodes are homogeneous, exhibiting low echogenicity and showing no areas of degeneration or necrosis, representing the classic homogenous “black hole sign”.

The margins. The blurring of lymph node borders, or in some extreme cases, evident infiltration into adjacent tissues, is typically indicative of a malignant process. However, it should be remembered that in cases of inflammatory changes, the infiltration may also extend beyond the capsule, leading to abscess formation or even cutaneous fistulas. Additionally, the assessment of lymph node margins via ultrasound is somewhat challenging due to the physical principles of ultrasonography. With high-frequency probes and superficially located nodes, it is possible to obtain highly detailed images of the lymph node borders. However, these are significantly better visualized in areas where the node capsule is parallel to the probe surface and perpendicular to the ultrasound beam. In regions where the capsule aligns with the probe axis, an apparent blurring effect can occur, mimicking boundary irregularities. Therefore, while blurring of lymph node borders is a feature that warrants vigilance, it is essential to consider that it may result from an inflammatory process. On the other hand, most malignant lymph nodes retain a smooth capsule.

The vascular architecture of lymph nodes is an important feature supporting the differentiation between benign and malignant processes. We identified five types of vascularisation. 1. no visible vascularization, 2. slight vascularization within the sinus (small tree), 3. vascularization extending slightly beyond the lymph node sinus (big tree) but not reaching the capsule. 4. peripheral, and 5. chaotic. The absence of visible vascularization does not rule out malignancy in a lymph node, as it can result from several factors. The node may be necrotic, due to systemic therapy such as anti-angiogenic drugs or following radiotherapy, or a specific tumour may have low angiogenic potential, resulting in a limited vascular network. The lack of visible vessels may also be due to technical limitations; deeper lymph nodes require different imaging parameters, often reducing the sensitivity of color Doppler. In the case of nodes adjacent to large vessels, motion artifacts may mimic or disrupt flow signals within the node. Reducing probe sensitivity to eliminate artifacts may also lead to the exclusion of true vascular flows within the lymph node. In general, the first three patterns are typically observed in benign LNs; however, they do not have exclusionary value, as they can also be found in neoplastic nodes. In the case of the fourth peripheral vascularization, two variants of subcapsular vessels are possible. In one variant, the vessels are confined exclusively to the subcapsular region, while in the other variant, they extend from the sinus through the cortex, reaching the capsule. In both variants, the presence of subcapsular vessels (peripheral type) correlates with an increased risk of malignancy. In the fifth type, the vessels are chaotic, reflecting neoangiogenesis, which serves as the strongest predictor of malignancy. Our observations concerning patterns of vasculature (Table 5) align with studies by other authors [42,43,44,45,46], where the basis for identifying malignant lymph nodes is increased vascularization outside the hilum, both peripheral (type 4) and chaotic (type 5). Type 4 vascularization is more commonly observed in systemic diseases, whereas chaotic type 5 vascularization is more frequently associated with metastatic lymph nodes. Enhanced hilar vascularization (type 3) is typically seen in reactive lymph nodes, while minimal hilar vascularization (type 2) is characteristic of normal lymph nodes.

However, deviations from general patterns should be considered, particularly since absent or weak vascularization can also be encountered in metastatic lymph nodes, and therefore, types 1 or 2 do not exclude a neoplastic process.

Elastography. Numerous publications and meta-analyses highlight the significant value of elastography in lymph node diagnostics [47,48,49,50,51,52,53]. Elastography can assist in differentiating malignant lymph nodes from benign ones. Most results indicate lower elasticity in neoplastic nodes compared to normal or reactive lymph nodes. However, noticeable differences exist between studies due to varying techniques (strain or shear wave), equipment model/manufacturer, types of cancers, and criteria, which involve a considerable degree of subjectivity. One of the large meta-analyses on elastography, involving 1411 axillary lymph nodes in breast cancer, showed high accuracy, with an overall sensitivity of 0.79, specificity of 0.90, and an AUC of 0.91 [50]. In our study, only a small group of patients (n = 42) was tested using elastography, so we did not statistically assess the efficacy of this technique. Based on numerous studies by other authors, it can be inferred that elastography is a valuable option in ultrasound, which could complement the array of multiparametric assessments and improve the accuracy of LN-RADS classification.

### 4.2. Limitations

This study was conducted retrospectively at a single institution on a moderate number of LNs. Moreover, no formal statistical comparison between genders or anatomical regions was performed in this study. However, we believe that the sample size of 719 LNs provides a robust foundation for the initial validation of the LN-RADS system. Furthermore, our primary objective was to present the LN-RADS system and validate its diagnostic performance on a large and diverse group of superficial LNs. We intentionally focused on developing and testing the system across a broad spectrum of cases without stratifying results by factors such as gender or anatomical location, as these were not the central variables of interest in our analysis. Further prospective studies with larger and controlled patient cohorts are necessary to confirm our findings.

Second, the evaluation performed by all three radiologists was based solely on data acquired by a senior consultant. We suspect that less experienced radiologists may require additional training in LN visualization techniques and in obtaining all the key parameters necessary for the LN-RADS system.

Additionally, this study focused exclusively on superficial lymph nodes located in the neck, peri-clavicular region, axilla, and inguinal region. Applying the LN-RADS system to deep-seated lymph nodes, such as those found in the abdomen and pelvis, would be significantly more challenging due to differences in imaging modalities and anatomical complexity. However, the aim of this article was to address superficial lymph nodes specifically. In future work, we plan to expand the LN-RADS scale to enable its application in CT and MRI imaging for lymph nodes located in other anatomical regions.

Finally, the strongest advantage of the LN-RADS system—its flexibility—might also be considered its greatest weakness. Nowadays, medical professionals are accustomed to direct and rigid guidelines as well as standardized diagnostic algorithms. In contrast, the LN-RADS system offers a holistic approach, allowing for individualized assessment based on a broad range of radiological and clinical data. While this flexibility enhances adaptability to complex cases, it may also introduce variability in interpretation, requiring a higher level of expertise and experience.

## 5. Summary

The LN-RADS system demonstrated high effectiveness in differentiating benign from malignant LNs on ultrasound (US), enabling precise risk stratification across defined groups of patients. It proved successful in evaluating all LNs within the entire study cohort and in specific subgroups, thanks to its open and flexible assessment model. The system is straightforward and intuitive, relying on heuristic evaluation. It integrates all available data, including morphological LN characteristics and clinical information, without imposing restrictions on the number of assessed features or rigid rules. This flexibility enables the application of the most relevant discriminative predictors in various clinical scenarios.

The substantial agreement between three researchers with varying levels of expertise indicates that LN-RADS can be widely adopted without requiring extensive experience or expert knowledge. The system was tested in a large multidisciplinary regional hospital using an unselected dataset, which included oncologic, hematologic, reactive, and healthy lymph nodes. Healthy nodes served as an excellent control group. These results suggest that the proposed system is suitable for implementation in routine clinical practice.

In conclusion, LN-RADS enables the differentiation of malignant and benign superficial LNs on US with high diagnostic accuracy. The high agreement between an experienced expert and a briefly trained resident underscores its reproducibility, making it a practical tool for daily LN metastasis risk stratification. This system has the potential to streamline clinical decision making and enhance patient management.

## Figures and Tables

**Figure 1 cancers-17-02030-f001:**
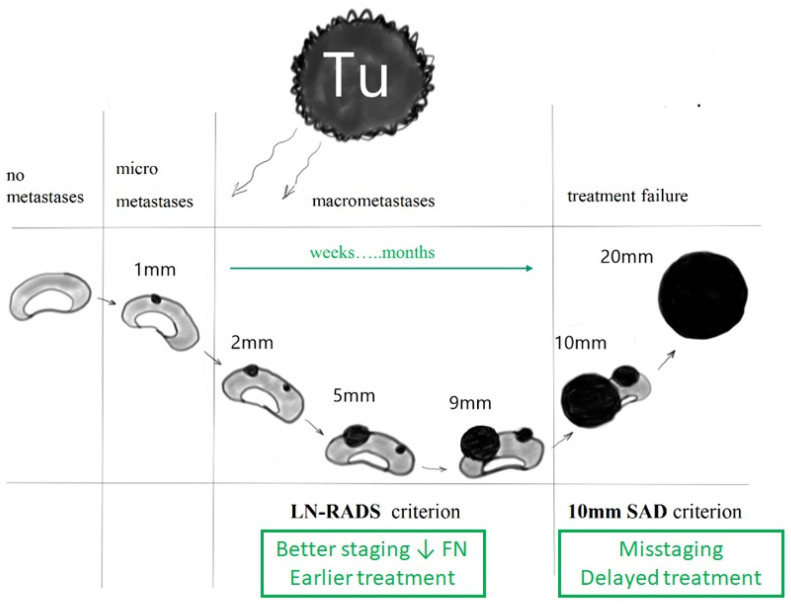
The pattern of development of lymph node metastases is usually similar regardless of the type of cancer and the location of the LN—initially micrometastases, gradual growth to macrometastases, merging of foci into larger infiltrates, and in the final stage, involvement of the entire node, often with concomitant retrograde changes like necrosis.

**Figure 2 cancers-17-02030-f002:**
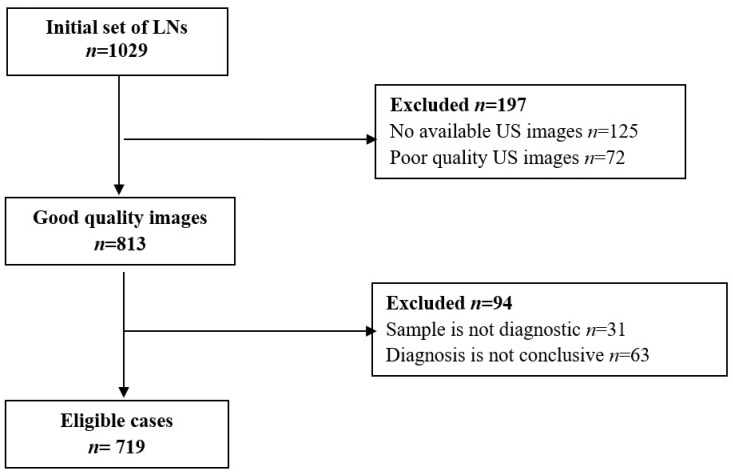
Flowchart of LNs’ inclusion in the study.

**Figure 3 cancers-17-02030-f003:**
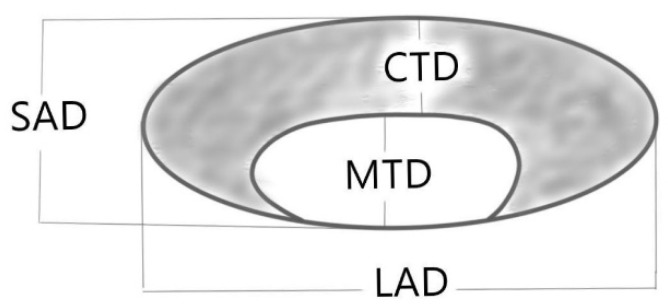
Objective morphological features of LNs.

**Figure 4 cancers-17-02030-f004:**
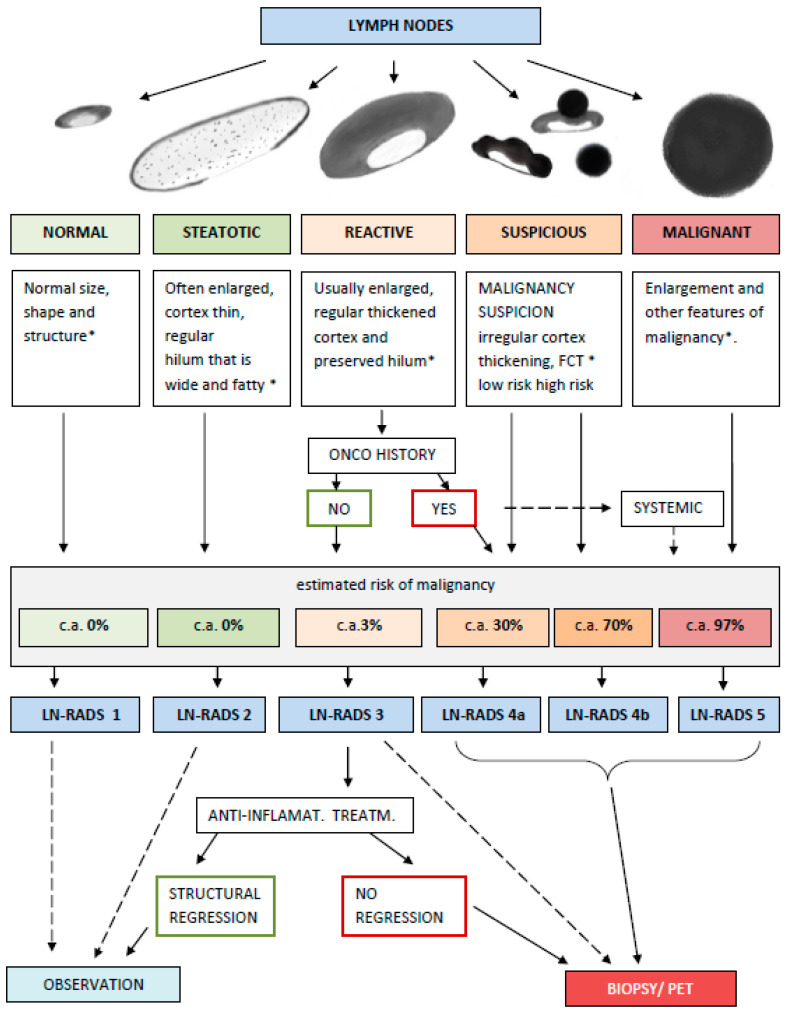
Summarized application of LN-RADS system in practice. * More details on features are located in the main text. C.a.—risk of cancer.

**Table 1 cancers-17-02030-t001:** Subjective morphological features of LNs.

	Grid	1	2	3	4	5
Feature	
Shape	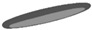	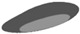	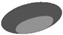	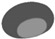	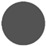
Cortex Irregularity	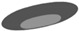	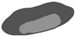	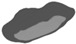	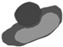	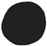
CortexEchogenicity	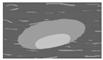	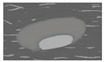	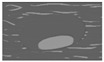	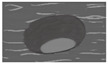	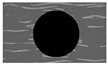
CortexInhomogenity	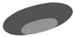	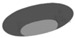	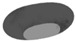	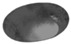	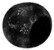
Borders	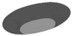	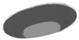	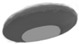	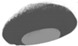	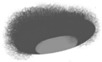
Vascularpattern	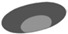	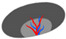	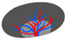	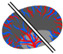	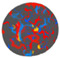

Note: The 5-point scale for assessing subjective features is not directly related to the LN-RADS scale. Each feature must be considered individually and independently, as detailed in the “Discussion” chapter. Detailed description of above features is provided in Appendix A.

**Table 2 cancers-17-02030-t002:** The number of true-negative (TN) and false-negative (FN) diagnoses of malignant lymph nodes for LN-RADS-1, LN-RADS-2, LN-RADS-3, and LN-RADS-4a categories, as well as the number of true-positive (TP) and false-positive (FP) diagnoses of malignant lymph nodes for LN-RADS-4b and LN-RADS-5 categories.

Benign			Malignant		
Category	TN	FN	Category	TP	FP
LN-RADS 1	33	0	LN-RADS 4b	142	42
LN-RADS 2	46	0	LN-RADS 5	204	7
LN-RADS 3	107	2	Sum	346	49
LN-RADS 4a	95	41			
Sum	281	43			

**Table 5 cancers-17-02030-t005:** Types of vascular architecture (analyzed in 238 LNs using Color Doppler US) with histopathological correlations.

Vascular Pattern TypeDescription	Scheme	Malignancy Probability	Benignin HP	Malignantin HP
Vascular-grid-1No visible blood flow	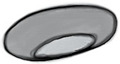	53%	8	9
Vascular-grid-2Hilar flow(small tree)	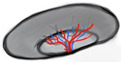	29%	36	15
Vascular-grid-3 Hilar-cortical flow (big tree)	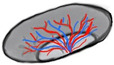	54%	11	13
Vascular-grid-4Peripheral vasculature	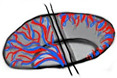	78%	17	60
Vascular-grid-5Chaotic vascular architecture	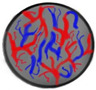	92%	7	62

## Data Availability

Data supporting the reported results are available upon request.

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
