# Peer review of "Lymph Node Reporting and Data System (LN-RADS)—Retrospective Evaluation for Ultrasound Classification of Superficial Lymph Nodes"

_cancers, 2025, doi:10.3390/cancers17122030_

Round 1
Reviewer 1 Report
Comments and Suggestions for Authors
In general, I liked the manuscript. I found it relatively easy to read and understand. I especially liked how it was written in a more narrative format than a highly structured technical format. I would like to suggest some ways to improve the manuscript.
Inter-Observer Agreement (lines 396-400). After each Cohen's Kappa, please remove the p-value and replace it with a 95% confidence interval. In this context, the confidence interval surrounding each Kappa value should be much more informative scientifically than the Kappa's p-value.
In the same paragraph, line 398 says "trained resident (C)" while line 399 says "resident (CC)". Why is the resident given two different labels? Are there two different residents?
Table 4 is not annotated well. In each histology column, there are two numbers, the second of which is inside parentheses. Please tell the reader whether the first number is a mean or a median or some third thing. Please tell the reader whether the second number is a standard deviation, a standard error, or another third thing. In the last column, p-values are given. What test was used to generate the p-values, and what statistical null hypothesis was being tested?
Lines 219-221, the sentence that begins, "The system aims to detect...". That sentence is more of a goal than a method. Perhaps it should be moved to either the Introduction or the Discussion. In fact, I wonder whether lines 214-219 belong better in the Discussion, but I hesitate to suggest moving that passage because moving it may interrupt the narrative flow.
Lines 298-299, the sentence that says "The cut-off points were determined using the Youden criterion." That is too vague. What is the Youden criterion? Are the authors trying to say that the cut-off points were determined by maximizing the Youden Index and/or Youden's J-statistic?
Comments on the Quality of English Language98% of the English was excellent. But I saw some English-language things that need attention.
Line 41. "multiparametric" is misspelled.
Lines 210-214. Those lines contain a single sentence that has some garbled English in it. Please break it up into two sentences, and then please have someone good at written English review the result.
Line 225. Change "according" to "accordingly".
Lines 284 and 285. Change the spelling of "morfological" to "morphological".
Line 294. The "Manny-Whitney test" is misspelled. Please change the spelling of "Manny" to "Mann".
Line 297. Please do not use Italics when naming the Chi-squared test.
Line 299. Please rewrite the beginning of the sentence to say, "In the aforementioned statistics, parameters...", i.e., please insert the word "the" and please put a comma after the word "statistics".
Lines 447-449, the part that begins, "rather than a single parametric model...". The English is garbled. I'm not sure what an exemplary 10mm SAD criterion is or what a closed few parametric calculation-based model means.
Line 453. Please change "eliminates" to "eliminate" so that its tense agrees with the tense of "compensate" in the previous line.
Author Response
Thank you very much for your careful reading of our manuscript and for your constructive and thoughtful comments. We truly appreciate the time and effort you invested in reviewing our work.
We are pleased that the manuscript was generally well received and that you appreciated its more narrative format. Below, we address each of your comments point by point. Where relevant, we have revised the text accordingly to improve its clarity and scientific precision.
Comment 1: Inter-Observer Agreement (lines 396-400). After each Cohen's Kappa, please remove the p-value and replace it with a 95% confidence interval. In this context, the confidence interval surrounding each Kappa value should be much more informative scientifically than the Kappa's p-value.
Response 1: Thank you very much for this valuable suggestion. We agree that reporting the 95% confidence intervals (CIs) provides more informative and scientifically relevant insight than p-values in the context of inter-observer agreement. We have removed the p-values and added 95% confidence intervals for each Cohen’s kappa value as suggested.
Revised text (lines 396–400):
Cohen’s kappa value between expert reader (A) and experienced reader (B) was 0.766 (95% CI: 0.729–0.802), between expert reader (A) and trained resident (C) was 0.686 (95% CI: 0.647–0.726), and between experienced reader (B) and resident (C) was 0.655 (95% CI: 0.614–0.695). Interobserver agreement in all analyzed cases was substantial.
Comment 2: In the same paragraph, line 398 says "trained resident (C)" while line 399 says "resident (CC)". Why is the resident given two different labels? Are there two different residents?
Response 2: Thank you for noticing this inconsistency. This was an unintentional error — both instances refer to the same individual, the trained resident. We have corrected the label to consistently use "(C)" throughout the paragraph to avoid confusion.
Comment 3: Table 4 is not annotated well. In each histology column, there are two numbers, the second of which is inside parentheses. Please tell the reader whether the first number is a mean or a median or some third thing. Please tell the reader whether the second number is a standard deviation, a standard error, or another third thing. In the last column, p-values are given. What test was used to generate the p-values, and what statistical null hypothesis was being tested?
Response 3: Thank you for your detailed and helpful comment. We acknowledge this lack of clarity in the current version of Table 4. Numbers represent dimentions given in milimeters. The first number in each cell represents the mean value, and the number in parentheses represents the corresponding standard deviation. We have added a note to the table legend to clearly indicate this. Regarding the p-values presented in the last column, they were calculated using the Kruskal–Wallis test, as the data did not meet the assumptions of normality. While we did not define a formal statistical null hypothesis in the manuscript, the implicit assumption was that there is no statistically significant difference in a given parameter across the different histological types of nodal metastases. Our goal in this analysis was to explore whether any parameter might differentiate between those histological subtypes. In addition, to improve clarity, we have expanded all abbreviations used in the table in a footnote below the table.
Comment 4: Lines 219–221, the sentence that begins, "The system aims to detect...". That sentence is more of a goal than a method. Perhaps it should be moved to either the Introduction or the Discussion. In fact, I wonder whether lines 214–219 belong better in the Discussion, but I hesitate to suggest moving that passage because moving it may interrupt the narrative flow.
Response 4: Thank you for this insightful comment. We agree that the sentence beginning with “The system aims to detect…” is more appropriately framed as a goal rather than a methodological detail. Following your suggestion, we have moved this sentence to the end of the Introduction, where it fits naturally as a summary of the motivation behind developing the LNRADS system. Regarding the preceding lines (214–219), we also agree with your concern — while they contain elements that could be moved to the Discussion, we believe that doing so would interrupt the narrative flow of the paper, especially in such a technical section as Materials and Methods. We therefore opted to retain their current placement to preserve clarity and cohesion in the manuscript’s structure.
Comment 5: Lines 298–299, the sentence that says "The cut-off points were determined using the Youden criterion." That is too vague. What is the Youden criterion? Are the authors trying to say that the cut-off points were determined by maximizing the Youden Index and/or Youden's J-statistic?
Response 5: Thank you for pointing out this lack of clarity. We apologize for the imprecise wording. Indeed, we determined the cut-off points by maximizing the Youden Index (Youden’s J-statistic). We have revised the sentence accordingly to make this explicit in the text.
Revised sentence (lines 298–299):
The cut-off points were determined by maximizing the Youden Index (Youden’s J-statistic).
Comments on the Quality of English Language:
98% of the English was excellent. But I saw some English-language things that need attention.
Response:
We sincerely thank you for your careful and constructive language suggestions, which have helped us significantly improve the clarity and quality of the manuscript. We have carefully addressed all the points you raised and made the necessary corrections, including:
-
Corrected spelling of multiparametric, morphological, and Mann–Whitney test.
-
Replaced “according” with “accordingly” (line 225).
-
Removed italics from “Chi-squared test” (line 297).
-
Revised the sentence beginning “In the aforementioned statistics…” (line 299) as requested.
-
Fixed subject–verb agreement between “eliminate” and “compensate” (line 453).
-
Rewritten the more garbled or unclear passages, particularly in lines 210–214 and 447–449, breaking up complex sentences and clarifying their meaning in standard scientific English.
Once again, thank you for your close reading and helpful suggestions — they have had a meaningful impact on the manuscript’s readability and precision.
Reviewer 2 Report
Comments and Suggestions for Authors
This manuscript discusses LN reporting and data system evaluation.
Major concerns and comments:
- Which lymph node is studied in this manuscript? Any lymph node in the human body? Please discuss.
- Please add the patients' detailed info, such as BMI/BRI, gender, age, types of tumors/cancers, etc.
- Please list all abbreviations.
- Does lymph node location matter in this study's conclusion?
- Any gender difference on this study's conclusion?
- On Table 3: any significant statistical analysis?
- On Figure 5: please use color line for each condition.
- On Table 4: for example, what does 17.2 mean and what does 10.0 mean?
Author Response
Dear Reviewer,
Thank you very much for your thorough and constructive review of our manuscript. We greatly appreciate your insightful comments, which have helped us to improve the clarity and quality of our work. Our detailed responses to your comments are provided below.
Comment 1: Which lymph node is studied in this manuscript? Any lymph node in the human body? Please discuss.
Response 1:
Thank you very much for this important observation. In the Materials and Methods section, we specified that samples were obtained from superficial lymph nodes of the neck, peri-clavicular region, axilla, and inguinal region (with corresponding sample sizes). Additionally, the title of the manuscript clearly indicates that the focus is on superficial lymph nodes. We believed this information made the scope clear; however, to avoid any potential ambiguity, we have now emphasized this point more explicitly in the Discussion, particularly in the Limitations section. We added a paragraph as follows:
Additionally, this study focused exclusively on superficial lymph nodes located in the neck, peri-clavicular region, axilla, and inguinal region. Applying the LN-RADS system to deep-seated lymph nodes, such as those found in the abdomen and pelvis, would be significantly more challenging due to differences in imaging modalities and anatomical complexity. However, the aim of this article was to address superficial lymph nodes specifically. In future work, we plan to expand the LN-RADS scale to enable its application in CT and MRI imaging for lymph nodes located in other anatomical regions.
Comment 2: Please add the patients' detailed info, such as BMI/BRI, gender, age, types of tumors/cancers, etc.
Response 2: Thank you for this comment. Detailed patient information including gender and age is provided in the Materials and Methods section, with additional details on tumor types presented in the Results section under the cohort analysis. Unfortunately, we do not have data on BMI or BRI available in our database, so these parameters could not be included in the study. Moreover, considering that the study focuses on superficial lymph nodes, we believe that BMI/BRI would not have a meaningful impact on the actual ability to visualize these lymph nodes.
Comment 3: Please list all abbreviations.
Response 3: Thank you for the valuable suggestion. We have compiled and included a comprehensive list of all abbreviations used in the manuscript for clarity and ease of reference. The list is as follows:
Abbreviations
- ADC – Apparent Diffusion Coefficient
- AUC – Area Under the Curve
- BI-RADS – Breast Imaging Reporting and Data System
- BMI – Body Mass Index
- BRI – Body Roundness Index
- CNB – Core Needle Biopsy
- CT – Computed Tomography
- DWI – Diffusion-Weighted Imaging
- FNB – Fine Needle Biopsy
- H-P – Histopathological
- LAD – Long Axis Diameter
- LN – Lymph Node
- LN-RADS – Lymph Node Reporting and Data System
- MRI – Magnetic Resonance Imaging
- SAD – Short Axis Diameter
- US – Ultrasound
- VAB – Vacuum-Assisted Biopsy
Comments 4 and 5: Does lymph node location matter in this study's conclusion? Any gender difference on this study's conclusion?
Responses 4 and 5:
Thank you for these relevant questions.
In this study, our primary objective was to present the LN-RADS system and validate its diagnostic performance on a large and diverse group of superficial lymph nodes (LNs). We intentionally focused on developing and testing the system across a broad spectrum of cases, without stratifying results by factors such as gender or anatomical location, as these were not the central variables of interest in our analysis.
Our rationale was that LN morphology, particularly in pathological nodes, is more strongly influenced by the type of underlying disease than by the patient’s gender or the exact anatomical location of the lymph node. This approach allowed us to test LN-RADS as a general diagnostic tool, applicable in routine clinical settings involving superficial LNs.
However, we acknowledge that no formal statistical comparison between genders or anatomical regions was performed in this study, and we have clarified this point in the revised version of the Discussion, particularly in the Limitations section.
Comment 6: On Table 3: any significant statistical analysis?
Response 6: Thank you for the suggestion. Table 3 presents diagnostic performance metrics (sensitivity, specificity, accuracy, PPV, NPV) for individual parameters in identifying malignant lymph nodes. These metrics are descriptive and were not subject to direct statistical comparison between each parameter. Therefore, no significance testing was applied in this table. The purpose of this analysis was to provide insight into the individual predictive performance of each morphological feature used in the LN-RADS system.
Comment 7: On Figure 5: please use color line for each condition.
Response 7: Thank you for your comment. We tested alternative versions of Figure 5 using color-coded arrows for each condition. However, due to the predominantly bright tones used in the image background, the application of multiple colored arrows significantly reduced the overall clarity and legibility of the figure. We believe that visual readability should remain a priority in this case. Of course, if the journal editor ultimately suggests the use of color-coded lines, we are open to adjusting the figure accordingly. We would prefer to follow the journal's guidelines or editorial decision on this matter.
Comment 8: On Table 4: for example, what does 17.2 mean and what does 10.0 mean?
Response 8: Thank you for pointing this out. You are absolutely right — the original table caption and legend were incomplete, which caused confusion. This was our oversight, and we have now corrected it by updating the table title and adding a proper legend to clearly explain the meaning of the values. These numbers represent measurement values expressed in millimeters.
Round 2
Reviewer 2 Report
Comments and Suggestions for Authors
The authors answered my questions and comments. No more comments.